# Large Language Models Meet Open-World Intent Discovery and Recognition: An Evaluation of ChatGPT

**Xiaoshuai Song**[1*], **Keqing He**[2*], **Pei Wang**[1], **Guanting Dong**[1], **Yutao Mou**[1]
**Jingang Wang**[2], **Yunsen Xian**[2], **Xunliang Cai**[2], **Weiran Xu**[1*]
[1]Beijing University of Posts and Telecommunications, Beijing, China
[2]Meituan, Beijing, China
{songxiaoshuai,wangpei,dongguanting,myt,xuweiran}@bupt.edu.cn
{hekeqing,wangjingang,xianyunsen,caixunliang}@meituan.com

## Abstract

The tasks of out-of-domain (OOD) intent discovery and generalized intent discovery (GID) aim to extend a closed intent classifier to open-world intent sets, which is crucial to task-oriented dialogue (TOD) systems. Previous methods address them by fine-tuning discriminative models. Recently, although some studies have been exploring the application of large language models (LLMs) represented by ChatGPT to various downstream tasks, it is still unclear for the ability of ChatGPT to discover and incrementally extent OOD intents. In this paper, we comprehensively evaluate ChatGPT on OOD intent discovery and GID, and then outline the strengths and weaknesses of ChatGPT. Overall, ChatGPT exhibits consistent advantages under zero-shot settings, but is still at a disadvantage compared to fine-tuned models. More deeply, through a series of analytical experiments, we summarize and discuss the challenges faced by LLMs including clustering, domain-specific understanding, and cross-domain in-context learning scenarios. Finally, we provide empirical guidance for future directions to address these challenges.[1]

## 1 Introduction

Traditional task-oriented dialogue (TOD) systems are based on the closed-set hypothesis (Chen et al., 2019; Yang et al., 2021; Zeng et al., 2022) and can only handle queries within a limited scope of in-domain (IND) intents. However, users may input queries with out-of-domain (OOD) intents in the real open world, which poses new challenges for TOD systems. Recently, a series of tasks targeting OOD queries have received extensive research. OOD intent discovery (Lin et al., 2020; Zhang et al., 2021; Mou et al., 2022a) aims to group OOD queries into different clusters based

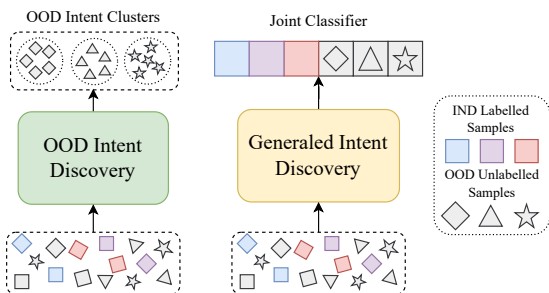

Figure 1: Illustration of OOD Intent Discovery and GID.

on their intents, which facilitates identifying potential directions and developing new skills. The General Intent Discovery (GID) task (Mou et al., 2022b) further considers the increment of OOD intents and aims to automatically discover and increment the intent classifier, thereby extending the scope of recognition from existing IND intent set to the open world, as shown in Fig 1.

Previous work studied above OOD tasks by fine-tuning the discriminative pre-training model BERT (Devlin et al., 2018). Recently, a series of powerful generative LLMs have been proposed one after another, such as GPT-3 (Brown et al., 2020), PaLM (Chowdhery et al., 2022) and LLaMA (Touvron et al., 2023). The emergence of LLMs has brought revolutionary changes to the field of natural language processing (NLP). Given the superior in-context learning ability, prompting LLMs has become a widely adopted paradigm for NLP research and applications (Dong et al., 2022b). Since these LLMs are trained on a large amount of general text corpus and have excellent generalization ability, this triggers the thinking of what benefits LLMs can bring and what challenges will LLMs face when applying them to open-scenario intent discovery and recognition?

As one of the representative LLMs, ChatGPT, developed by OpenAI, has attracted significant attention from researchers and practitioners in a short period of time. While the NLP community has

---

*The first two authors contribute equally. Weiran Xu is the corresponding author.

[1]We release our code at https://github.com/songxiaoshuai/OOD-Evaluation

been studying the ability of LLMs to be applied to various downstream tasks, such as translation (Jiao et al., 2023), mathematics (Frieder et al., 2023), education (Malinka et al., 2023),emotion recognition(Lei et al., 2023), its ability in OOD is still not fully explored. Different from Wang et al. (2023) which evaluate the robustness of ChatGPT from the adversarial and out-of-distribution perspective, OOD intent discovery and GID focuses on using IND knowledge transfer to help improve TOD systems with OOD data. In this paper, on the one hand, we focus on OOD intent discovery and GID the two open-scenario intent tasks to explore whether prompting ChatGPT can achieve good performance in discovering and incrementally recognizing OOD intents; on the other hand, we aim to gain insights into the challenges faced by LLMs in handling open-domain tasks and potential directions for improvement.

To the best of our knowledge, we are the first to comprehensively evaluate ChatGPT's performance on OOD intent discovery and GID. In detail, we first design three prompt-based methods based on different IND prior to guide ChatGPT to perform OOD discovery in an end-to-end manner. For GID, we innovatively propose a pipeline framework for performing GID task under a generative LLM (Section 3). Then we conduct detailed comparative experiments between ChatGPT and representative baselines under three dataset partitions (Section 4). In order to further explore the underlying reasons behind the experiments, we conduct a series of analytical experiments, including in-context learning under cross-domain demonstrations, recall analysis, and factors that affect the performance of ChatGPT on OOD discovery and GID. Finally, we compare the performance of different LLMs on these OOD tasks (Section 5).

**Our findings.** The major findings of the study include:

**What ChatGPT does well:**

- ChatGPT can perform far better than the non-fine-tuned BERT on OOD tasks without any IND prior, thanks to its powerful semantic understanding ability.

- For OOD intent discovery, when there are few samples for clustering, ChatGPT's performance can rival that of fine-tuned baselines.

- ChatGPT can simultaneously perform text clustering and induce the intent of each cluster, which is not available in the discriminant model.

**What ChatGPT does not do well:**

- For OOD intent discovery, ChatGPT performs far worse than the fine-tuned baselines under multi-sample or multi-category scenes, and is severely affected by the number of clusters and samples, with poor robustness.

- For GID, the overall performance of ChatGPT is inferior to that of the fine-tuned baselines. The main reason is the lack of domain knowledge, and the secondary reason is the quality of the pseudo-intent set.

- There are obvious recall errors in both OOD discovery and GID. In OOD discovery, this is mainly due to the generative architecture of ChatGPT. In GID, recall errors are mainly caused by ChatGPT's lack of domain knowledge and unclear understanding of intent set boundaries.

- ChatGPT can hardly learn knowledge from IND demonstrations that helps OOD tasks and may treat IND demonstrations as noise, which brings negative effects to OOD tasks.

In addition to the above findings, we further summarize and discuss the challenging scenarios faced by LLMs including **large scale clustering, semantic understanding of specific domain and cross-domain in-context learning** in Section 6 as well as provide guidance for future directions.

## 2 Related Work

### 2.1 Large Language Models

Recently, there are growing interest in leveraging large language models (LLMs) to perform various NLP tasks, especially in evaluating ChatGPT in various aspects. For example, Frieder et al. (2023) investigate the mathematical capabilities of ChatGPT by testing it on publicly available datasets, as well as hand-crafted ones. Tan et al. (2023) explore the performance of ChatGPT on knowledge-based question answering (KBQA). Wang et al. (2023) evaluate the robustness of ChatGPT from the out-of-distribution (OOD) perspective. A series works by Liu et al. (2023); Guo et al. (2023); Dong et al. (2022a, 2023) explore the impact of input perturbation problems on model performance

Next, I will first give you a set of sentences , which will be recorded as Set 1. First,Please classify the sentences in Set 1 into 5 (Number of Clusters) categories according to their intentions. You only need to output the category number and the corresponding sentence number in the following format:
Category 1: 1,2,3,4,5 ……
Set 1($D_{test}^{OOD}$): 1. can I choose a date for delivery? ……
**Direct clustering (DC)**

Next, I will first give you a set of intent categories, which will be recorded as Set 1. Then I will give you another set of sentences without intention labels, recorded as Set 2. You first need to learn the knowledge in Set 1, and then use the learned knowledge to classify the sentences in Set 2 into 5 (Number of Clusters) categories according to their intentions. You only need to output the category number and the corresponding sentence number in the following format:
Category 1:  1,2,3,4,5 ……
It should be noted that the intention in set 1 and the intention in set 2 do not overlap.
Set 1($Y^{IND}$): pin_blocked; ……
Set 2($D_{test}^{OOD}$): 1. can I choose a date for delivery? ……
**Zero Shot Discovery (ZSD)**

Next, I will first give you a set of sentences with intention labels, which will be recorded as Set 1.Then I will give you another set of sentences without intention labels, recorded as Set 2. You first need to learn the knowledge in Set 1, and then use the learned knowledge to classify the sentences in Set 2 into 5 (Number of Clusters) categories according to their intentions. You only need to output the category number and the corresponding sentence number in the following format:
Category 1:  1,2,3,4,5 ……
It should be noted that the intention in set 1 and the intention in set 2 do not overlap.
Set 1 ($D_{demo}^{IND}$): How do I unblock my card?    intention:pin_blocked; ……
Set 2 ($D_{test}^{OOD}$): 1. can I choose a date for delivery? ……
**Few Shot Discovery (FSD)**

Figure 2: The different prompts for three methods of OOD intent discovery.

from small models to LLMs. In this paper, we aim to investigate the ability of ChatGPT to discover and increment OOD intents and further explore the challenges faced by LLMs and potential directions for improvement.

## 2.2 OOD Intent Discovery

Unlike the simple text clustering task, OOD intent discovery considers how to facilitate the discovery of unknown OOD intents using prior knowledge of IND intents . Lin et al. (2020) use OOD representations to compute similarities as weak supervision signals. Zhang et al. (2021) propose an iterative method, DeepAligned, that performs representation learning and clustering assignment iteratively while Mou et al. (2022c) perform contrastive clustering to jointly learn representations and clustering assignments. In this paper, we evaluate the performance of methods based ChatGPT about OOD discovery and provide a detailed qualitative analysis.

## 2.3 General Intent Discovery

Since OOD intent discovery ignores the fusion of IND and OOD intents, it cannot further expand the recognition range of existing TOD systems. Inspired by the above problem, Mou et al. (2022b) propose the General Intent Discovery (GID) task, which requires the system to discover semantic concepts from unlabeld OOD data and then jointly classifying IND and OOD intents automatically. Furthermore, Mou et al. (2022b) proposes two frameworks for performing GID task under discriminative models: pipeline-based and end-to-end frameworks. In this paper, we propose a new GID pipeline under generative LLMs and explore the performance of ChatGPT in different scenarios.

## 3 Methodology

### 3.1 Problem Formulation

**OOD Intent Discovery** Given a set of labeled IND dataset $D^{IND} = \{(x_i^{IND}, y_i^{IND})\}_{i=1}^{n}$ and unlabeled OOD dataset $D^{OOD} = \{(x_i^{OOD})\}_{i=1}^{m}$, where all queries from $D^{IND}$ belong to a predefined intent set $Y^{IND}$ containing $N$ intents, and all queries from $D^{OOD}$ belong to an unknown set $Y^{OOD}$ containing $M$ intents[2]. OOD intent discovery aims to cluster $M$ OOD groups from $D^{OOD}$ under the transfer of IND prior from $D^{IND}$.

**General Intent Discovery** GID aims to train a network that can simultaneously classify a set of labeled IND intent classes $Y^{IND}$ containing $N$ intents and discover new intent set $Y^{OOD}$ containing $M$ intents from an unlabeled OOD set $D^{OOD} = \{(x_i^{OOD})\}_{i=1}^{m}$. Unlike OOD discovery clustering that obtains $M$ OOD groups, the ultimate goal of GID is to expand the network's classification capability of intent query to the total label set $Y = Y^{IND} \cup Y^{OOD}$ containing $N + M$ intents.

### 3.2 ChatGPT for OOD Discovery

We evaluate the performance of ChatGPT on OOD intent discovery by designing prompts that include task instructions, test samples, and IND prior. We heuristically propose the following three methods based on different IND prior:

**Direct clustering (DC):** Since OOD intent discovery is essentially a clustering task, a naive approach is to cluster directly without utilizing any IND prior. The prompt is in the following

---

[2]Estimating $M$ is out of the scope of this paper. In the following experiment, we assume that $M$ is ground-truth and provide an analysis in Section 5.5. It should be noted that the specific semantics of intents in $Y^{OOD}$are unknown.

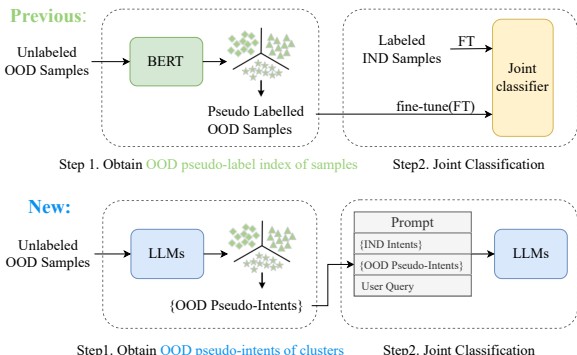

Figure 3: Comparison of the discriminant and generative GID framework.

format: <Cluster Instruction><Number of Clusters><Response Format><$D_{test}^{OOD}$>.

**Zero Shot Discovery (ZSD):** This method provides the IND intent set in the prompt as prior knowledge, but does not provide any IND samples. It can be used in scenarios where user privacy needs to be protected. The prompt is in the following format: <Prior: $Y^{IND}$><Cluster Instruction><Number of Clusters><Response Format><$D_{test}^{OOD}$>.

**Few Shot Discovery (FSD):** FSD provides several labelled samples for each IND intent in the prompt, hoping that ChatGPT can mine domain knowledge from IND demonstration and transfer it to assist OOD intent clustering. The prompt is in the following format: <Prior: $D_{demo}^{IND}$ & $Y^{IND}$><Cluster Instruction><Number of Clusters><Response Format><$D_{test}^{OOD}$>.

According to the input, ChatGPT outputs the index of OOD samples contained in each cluster, in the form of <Cluster Index><OOD Sample Index>. We show the prompts of these methods in Fig 2.

### 3.3 ChatGPT for GID

Previous discriminative GID framework first assign a pseudo-label index to each OOD sample through clustering and then jointly training the classifier with labeled IND data. However, the classification of queries by generative LLMs depends on specific intent semantics rather than abstract pseudo-label index symbols. Based on this, we innovatively propose a new framework that is suitable for generative LLMs, which relies on LLMs to generate an intent description with specific semantics as the pseudo-intent for each cluster, as shown in Fig 3.

In the first stage, on the basis of OOD intent discovery prompts, we add an additional instruc-

tion for generating intent descriptions, which are formally <OOD Discovery Prompt><Intent Describe Instruction>, input to ChatGPT, and obtain the intent description of each cluster. By aggregating these intent descriptions, we obtain the OOD pseudo-intent set $Y_{pseudo}^{OOD}$. Then, we incrementally add the pseudo-intent set to the existing IND intent set, i.e., $Y^{joint} = Y^{IND} \cup Y_{pseudo}^{OOD}$.

Next, we input prompts for joint IND and OOD intent classification in the following form: <Classification Instruction><Intent Set: $Y^{joint} >< x|x \in D_{test}^{IND} \cup D_{test}^{OOD} >$. According to the three different OOD discovery methods in Section 3.2, there are also three GID methods for ChatGPT: **GID under Direct clustering (GID-DC)**, **GID under Zero Shot Discovery (GID-ZSD)**, and **GID under Few Shot Discovery (GID-FSD)**. It should be noted that in the GID-FSD method, we provide few labeled samples of IND as demonstrations. The complete prompts are provided in Appendix D.

## 4 Experiment

### 4.1 Datasets

We conduct experiments on the widely used intent dataset Banking (Casanueva et al., 2020). Banking contains 13,083 user queries with 77 intents in the banking domain. Due to the length limitation of ChatGPT's conversations, we randomly sample 15 categories from Banking as IND intents and considered three OOD category quantity settings. Specifically, the OOD category quantity is 5 (IND/OOD=3:1), 10 (IND/OOD=3:2), and 15 (IND/OOD=1:1), respectively. For OOD intent discovery, we randomly sample 5 queries from the test set for each OOD class, where the ground-truth number of OOD classes is given as a prior. For GID, we randomly sample 10 queries from the test set for testing. In addition, only when using FSD or GID-FSD method, we randomly sample 3 queries for each IND class from the training set for demonstration. We provide detailed statistics of datasets in Appendix A.

### 4.2 Baselines

For OOD intent discovery, we choose to use BERT directly for k-means clustering (MacQueen, 1965) and two representative fine-tuned methods:

- **DeepAligned** (Zhang et al., 2021) is an improved version of DeepCluster(Caron et al., 2018). It designed a pseudo label alignment

| Prior | Method | IND/OOD=3:1 | | | IND/OOD=3:2 | | | IND/OOD=1:1 | | |
|---|---|---|---|---|---|---|---|---|---|---|
| | | ACC | NMI | ARI | ACC | NMI | ARI | ACC | NMI | ARI |
| W/O IND Prior | BERT | 52.00 | 41.58 | 15.36 | 36.00 | 42.33 | 5.616 | 29.33 | 45.26 | 2.499 |
| | ChatGPT(DC) | 88.00 | 84.62 | 73.36 | 78.00 | 78.20 | 55.32 | 58.22 | 65.30 | 28.21 |
| With IND Prior | DeepAligned | **100.0** | **100.0** | **100.0** | 78.67 | 82.18 | 61.01 | 74.67 | 80.83 | 55.35 |
| | DKT | 93.33 | 91.71 | 84.82 | **80.67** | **82.59** | **64.53** | **76.45** | **83.23** | **60.92** |
| | ChatGPT(ZSD) | 92.00 | 87.25 | 80.16 | 67.33 | 68.72 | 39.32 | 50.67 | 60.73 | 21.25 |
| | ChatGPT(FSD) | 74.67 | 64.77 | 45.92 | 56.67 | 63.56 | 31.47 | 49.78 | 60.98 | 20.72 |

Table 1: Performance comparison on OOD intent discovery. Results are averaged over three random run (p < 0.01 under t-test).

| Method | IND/OOD=3:1 | | | | | | IND/OOD=3:2 | | | | | | IND/OOD=1:1 | | | | | |
|---|---|---|---|---|---|---|---|---|---|---|---|---|---|---|---|---|---|---|
| | IND | | OOD | | ALL | | IND | | OOD | | ALL | | IND | | OOD | | ALL | |
| | F1 | ACC | F1 | ACC | F1 | ACC | F1 | ACC | F1 | ACC | F1 | ACC | F1 | ACC | F1 | ACC | F1 | ACC |
| DeepAligned-GID | 95.36 | 94.50 | **97.00** | **97.00** | 95.77 | 95.12 | 94.49 | 92.67 | **85.99** | **86.00** | 91.09 | 90.00 | 94.16 | 91.50 | **76.38** | **77.33** | 85.27 | 84.42 |
| E2E | **96.13** | **95.50** | **97.00** | **97.00** | **96.35** | **95.88** | **95.21** | **93.33** | 78.69 | 80.5 | 88.602 | 88.2 | **94.22** | **92.17** | 71.92 | 74.00 | 83.07 | 83.08 |
| ChatGPT(GID-DC) | 67.41 | 68.44 | 70.26 | 75.33 | 67.96 | 70.17 | 62.15 | 64.67 | 57.53 | 61.33 | 60.30 | 63.33 | 63.06 | 66.44 | 59.72 | 62.00 | 61.39 | 64.22 |
| ChatGPT(GID-ZSD) | 64.47 | 65.11 | 61.50 | 70.00 | 63.73 | 66.33 | 55.14 | 58.22 | 46.83 | 50.33 | 51.81 | 55.07 | 53.94 | 57.78 | 52.20 | 57.57 | 53.07 | 57.67 |
| ChatGPT(GID-FSD) | 72.77 | 79.11 | 20.27 | 17.33 | 59.65 | 63.67 | 68.74 | 74.89 | 50.75 | 52.00 | 61.54 | 65.73 | 68.29 | 74.89 | 52.57 | 51.78 | 60.43 | 63.33 |

Table 2: Performance comparison on GID.

strategy to produce aligned cluster assignments for better representation learning.

- **DKT** (Mou et al., 2022c) designs a unified multi-head contrastive learning framework to match the IND pretraining objectives and the OOD clustering objectives. In the IND pretraining stage, the CE and SCL objective functions are jointly optimized, and in the OOD clustering stage, instance-level CL and cluster-level CL objectives are used to jointly learn representation and cluster assignment.

For GID, the baselines are as follows:

- **DeepAligned-GID** is a representative pipeline method constructed by (Mou et al., 2022b) based on DeepAligned, which first uses the clustering algorithm DeepAligned to cluster OOD data and obtains pseudo OOD labels, and then trains a new classifier together with IND data.

- **E2E** (Mou et al., 2022b) mixes IND and OOD data in the training process and simultaneously learns pseudo OOD cluster as signments and classifies all classes via self-labeling. Given an input query, E2E connects the encoder output through two independent projection layers, IND head and OOD head, as the final logit and optimize the model through the unified classification loss, where the OOD pseudo label is obtained through swapped prediction (Caron et al., 2020).

We only use the samples belonging to IND and OOD intents in training set of Banking to train all fine-tuned methods.

### 4.3 Evaluation Metrics

For OOD intent discovery, We adopt three widely used metrics to evaluate the clustering results: Accuracy (ACC)[3], Normalized Mutual Information (NMI), and Adjusted Rand Index (ARI). For GID, we adopt two metrics: Accuracy (ACC) and F1-score (F1), to assess the performance of the joint classification results. Besides, we observe that all ChatGPT methods have the phenomenon that some samples are not assigned any cluster or intent (missing recall), while some are assigned multiple clusters or intents (repeated recall). For samples with missing recall, we randomly assign a cluster or intent; for those with repeated recall, we only retain the first assigned cluster or intent. We provide detailed recall analysis in Section 5.3.

### 4.4 Main Results

Table 1 and Table 2 respectively show the main results of the ChatGPT methods and baselines under OOD discovery and GID for three dataset divisions. Next, we analyze the results from three aspects:

(1) **Compare Method without IND Prior** From Table 1, we can see that without any IND prior knowledge, i.e., direct clustering, ChatGPT's performance is significantly better than BERT under three dataset partitions, indicating ChatGPT's

---

[3]We use the Hungarian algorithm (Kuhn, 1955) to obtain the mapping between the prediction and ground truth classes.

strengths in natural language understanding without using any private specific data. For example, under IND/OOD=3:1, ChatGPT(DC) outperforms BERT by 36.00% (ACC). As the number of OOD classes increases, the clustering metrics of Chat-GPT(DC) and BERT decrease rapidly, but Chat-GPT(DC) still achieves better performance than BERT, exceeding BERT by 28.89% (ACC) under IND/OOD=1:1.

(2) **Compare ChatGPT with Finetuned BERT** For OOD discovery, when the OOD ratio is relatively low, the optimal ChatGPT method is slightly inferior to fine-tuned baselines. However, as the OOD ratio increases, ChatGPT is significantly lower than fine-tuned model. We believe this is because as the OOD ratio increases, the number of clustered samples increases and more data brings more difficult semantic understanding challenges to generative LLMs. However, discriminative fine-tuned methods encode the samples one by one and are therefore less affected by the OOD ratio.

For GID, ChatGPT is significantly weaker than fine-tuned model in both IND and OOD metrics. According to Table 2, on average in three scenarios, the optimal ChatGPT method is weaker than the optimal fine-tuned method by 17.37% (IND ACC), 20.56% (OOD ACC), and 23.40% (ALL ACC), respectively. We believe this is because ChatGPT is pre-trained on large-scale general training data, which makes it difficult to perform better than fine-tuned models on specific domain data.

(3) **Compare different ChatGPT methods** For OOD discovery, DC generally achieves the best performance, while ZSD is slightly inferior, and FSD performs the worst. Although DC is slightly inferior to ZSD in the IND/OOD=3:1 scenario, it significantly outperforms other ChatGPT methods in the other two scenarios. FSD almost performs the worst among the three methods. ZSD provides additional prior knowledge of IND categories, while FSD provides labeled IND samples as context. However, more IND priors actually lead to worse performance for ChatGPT.

For GID, GID-FSD performs best on IND classification, while GID-DC performs best on OOD intents. Comparing GID-ZSD and GID-DC, the difference lies in the pseudo-intent set used. GID-ZSD is on average 6.22% (ALL ACC) behind GID-DC, indicating the importance of the pseudo-intent set. For GID-FSD, due to the IND demonstration, IND classification ability is significantly improved

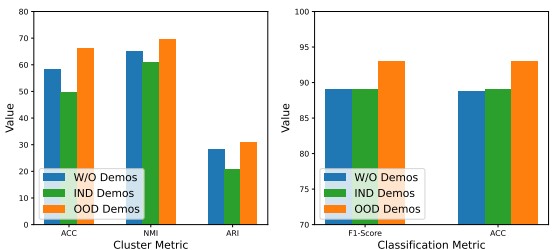

Figure 4: The impact of different demonstrations. W/O Demos means "no samples for demonstration". Under IND/OOD Demos, we demonstrate 3 labeled samples for each IND/OOD class.

through in-context learning. However, its OOD classification metric is not as good as that of GID-DC. We think this is because the quality of the pseudo-intent set induced by FSD is poor and the IND demonstration may be treated as noise. We leave the further analysis of demonstrations in Section 5.1 and GID exploration in Section 5.2.

# 5 Qualitative Analysis

## 5.1 In-context Learning from IND to OOD

In Section 4.4, we find that the IND prior does not bring positive effects to OOD tasks. To further explore the influence of different types of demonstrations, we compare the performance of OOD tasks under three demonstration strategies: W/O Demos, IND Demos, and OOD Demos. Specifically, for OOD discovery, we perform 15 OOD classes clustering; for classification, we test OOD classification under 10 classes, where the OOD intent used the ground-truth intent set to avoid the influence of pseudo intent set.

The results are reported in Fig 4. Compared with W/O Demos, OOD Demos achieve a significant performance improvement in both clustering and classification tasks. In contrast, IND Demos result in a performance decline in clustering and almost no improvement in classification. For OOD Demos, it can be considered that the demonstration and testing are of the same distribution, so ChatGPT can improve task performance through in-context learning. For IND Demos, the different distribution between demonstration and testing causes ChatGPT not only unable to bring performance gains through in-context learning but also regard demonstrations as in-context noise that interferes with task performance. **This shows that the distribution of demonstration text has a great impact on the effect of in-context learning**, as

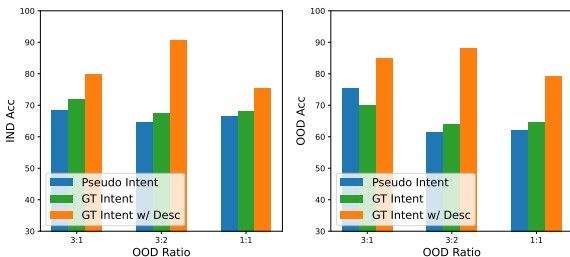

Figure 5: The impact of different intent sets on GID. The pseudo-intent set comes from ChatGPT(GID-SD).

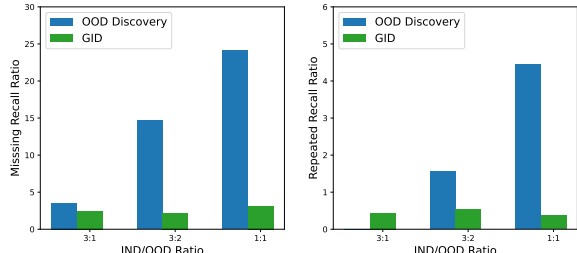

Figure 6: The incorrect recall ratio of ChatGPT. We average the statistics of three ChatGPT methods.

also mentioned in (Min et al., 2022). It should be noted that since both IND and OOD classes come from the banking domain, the fine-tuned model can improve OOD task performance through knowledge transfer from IND samples. However, this fails on ChatGPT, indicating that current in-context learning of LLMs lacks deep mining and transfer demonstration knowledge (i.e., from IND to OOD) capabilities.

## 5.2 Reasons for limiting GID performance

Since ChatGPT performs GID in a pipeline manner, we analyze ChatGPT's performance separately in generating pseudo intent sets and performing joint classification. We show a set of pseudo intent sets in Table 3. It can be seen that the three sets of pseudo intents are roughly similar to the real intents in semantics but show some randomness in granularity. Taking intent of ID 5 as an example, run-1 explains the payment method as "google pay or apple pay", which is consistent with the granularity of the real label; run-2 expands it to "different methods", and run-3 further broadens it to "payment related", with coarser granularity.

Next, we use the OOD ground-truth intent set to replace the OOD pseudo intent set and further add artificial descriptions for each IND&OOD intent, as shown in Fig 5. Compared with using pseudo intent set, using the ground truth intent set can only bring slight performance improvement, while adding intent descriptions can significantly improve classification performance. This shows that **the main reason for limiting ChatGPT's further improvement in GID task is the lack of domain knowledge, and the secondary reason is the quality of the pseudo intents**.

## 5.3 Recall Analysis

As mentioned in Section 4.3, ChatGPT has the problem of missing and repeated recall, which is a

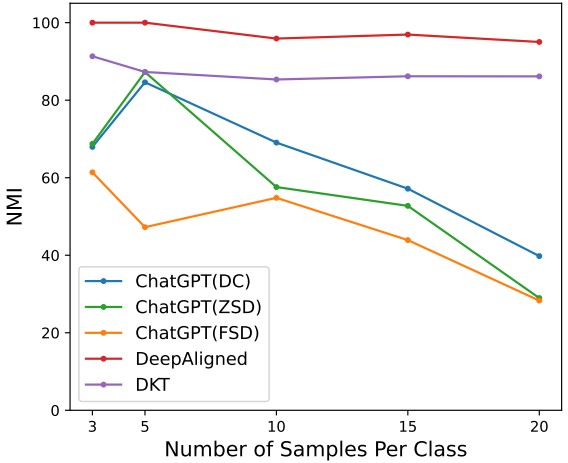

Figure 7: The impact of the number of samples in each cluster on OOD discovery under IND/OOD=3:1.

unique problem of generative LLMs.[4] Fig 6 shows the statistics of ChatGPT's incorrect recall. For OOD discovery, as the OOD ratio (clustering number) increases, the proportion of missing recall and repeated recall both increase significantly. For example, under IND/OOD=1:1, the probability of missing and repeated recall reaches 24.15% and 4.44% respectively, which has seriously damaged task performance. Since clustering tasks require inputting all samples into ChatGPT simultaneously, more samples bring more difficult task understanding and processing to ChatGPT, resulting in higher incorrect recall rates. For GID, the proportion of incorrect recall is almost unaffected by the OOD ratio, as GID is performed on a sample-by-sample basis. Furthermore, we find that incorrect recall on GID is mainly due to the lack of domain knowledge. This results in ChatGPT being unable to clearly identify intent set boundaries and may proactively allocate a query to multiple intents or refuse to allocate the query to predefined intent sets.

---

[4]We provide relevant cases in Appendix E.

| ID | Ground Truth Intent | Pseudo Intent 1 (run-1) | Pseudo Intent 2 (run-2) | Pseudo Intent 3 (run-3) |
|----|---------------------|-------------------------|-------------------------|-------------------------|
| 1 | card_delivery_estimate | Inquires about delivery time or schedule | Delivery and shipment related questions. | Delivery related inquiries |
| 2 | cancel_transfer | Requests for cancellation, urgent card needs, and transaction reversion | Transaction cancellation or reversion related questions. | Account/card related inquiries |
| 3 | verify_my_identity | Inquires about identity verification process | Identity verification related questions | Identity verification inquiries |
| 4 | cash_withdrawal_charge | Questions related to fees charged for transactions or withdrawals | Fee related questions, especially related to cash withdrawals. | Fee related inquiries |
| 5 | apple_pay_or_google_pay | Issues related to topping up accounts with mobile payment services such as Google Pay or Apple Pay. | Various questions related to top-ups and adding money to an account using different methods. | Payment related inquiries |

Table 3: Pseudo-intent set generated by three random runs of ChatGPT(GID-DC) under IND/OOD=3:1.

| Method | IND/OOD=3:1 | | IND/OOD=3:2 | | IND/OOD=1:1 | |
|--------|---------|-------|---------|-------|---------|-------|
| | K(Pred) | Error | K(Pred) | Error | K(Pred) | Error |
| DeepAligned($K^{'}$=2) | 4 | 1 | **11** | **1** | 11 | 4 |
| DeepAligned($K^{'}$=3) | 8 | 3 | 16 | 6 | **16** | **1** |
| ChatGPT(DC) | **5** | **0** | 14 | 4 | 23 | 8 |

Table 4: The results of estimating the number K of clusters, where $K^{'}$ is a hyperparameter for DeepAligned. The real number of clusters are 5,10,15, respectively.

| Model | OOD Discovery | | | GID | | |
|-------|------|------|------|-----------|-----------|-----------|
| | ACC | NMI | ARI | IND ACC | OOD ACC | ALL ACC |
| text-davinci-002 | 38.00 | 39.11 | 9.577 | 31.56 | 28.67 | 30.40 |
| text-davinci-003 | 71.33 | 72.12 | 49.84 | 59.56 | 63.33 | 61.07 |
| Claude | 70.00 | **84.27** | **62.24** | 56.67 | 70.00 | 62.00 |
| ChatGPT | **78.00** | 78.20 | 55.32 | **68.44** | **75.33** | **70.17** |

Table 5: Comparison of different LLMs. All LLMs use DC and GID-DC methods under IND/OOD=3:2.

## 5.4 Effect of Cluster Sample Number

We explore the effect of the number of clustered samples on ChatGPT by changing the ground-truth number of each OOD intent. As shown in Fig 7, **ChatGPT has poor robustness to the number of samples**. The clustering performance first reaches an optimal effect between 5 and 10 samples per class, and then drops rapidly. In contrast, discriminative fine-tuned methods exhibits good robustness. We believe this is because when there are too few samples, it's difficult for ChatGPT to discover clustering patterns. And when there are too many samples, ChatGPT needs to process too many samples at the same time, leading to more difficult clustering.

## 5.5 Estimate the Number of Cluster K

In the experiments above, the number of OOD classes was assumed to be ground truth. However, in real-world applications, the number of OOD clusters often needs to be estimated automatically. We use the same estimation algorithm DeepAligned as a baseline following (Zhang et al., 2021; Mou et al., 2022a). For ChatGPT, we remove the ground-truth cluster number from the prompt and count the estimated cluster number based on the result. The results are reported in Table 4. When the number of clusters is small, ChatGPT can obtain more accurate estimates. However, as the number of clusters increases, ChatGPT performs worse than the baseline and is prone to overestimate. For the baseline, an appropriate hy-

perparameter can achieve good results, but the robustness of the hyperparameter is poor. Therefore, **ChatGPT is more suitable for estimating a small number of clusters.**

## 5.6 Comparison of Different LLMs

In this section, we evaluate the performance of other mainstream LLMs and compare them with ChatGPT. Text-davinci-002 and text-davinci-003 belong to InstructGPT and text-davinci-003 is an improved version of text-davinci-002.[5] Compared with GPT-3, the biggest difference of Instruct-GPT is that it is fine-tuned for human instructions. In addition to the GPT family models, we also evaluate a new LLM Claude developed by Anthropic[6]. As shown in Table 5, ChatGPT performs better than text-davinci-002 and text-davinci-003 because ChatGPT is further optimized based on text-davinci-003. Claude shows competitive performance with ChatGPT on OOD discovery, but is weaker on GID. In addition, we try to evaluate GPT-3 (davinci) and find that GPT-3 fails to perform the tasks, which illustrates the importance of instruction tuning.

## 5.7 Effect of Different Prompt

Thorough prompt engineering is crucial to mitigate the variability introduced by different prompts. To address this, we devise three additional variations

---

[5]https://platform.openai.com/docs/models
[6]https://www.anthropic.com/product

| Prompt/Baseline | OOD Discovery | | | GID | | |
|---|---|---|---|---|---|---|
| | ACC | NMI | ARI | IND ACC | OOD ACC | ALL ACC |
| Original | 58.22 | 65.30 | 28.21 | 66.44 | 62.00 | 64.22 |
| Paraphrase | 58.67 | 66.80 | 30.10 | 69.33 | 62.00 | 65.67 |
| verbosity | 60.89 | 68.74 | 34.85 | 68.67 | 60.67 | 64.67 |
| Simplification | 53.78 | 64.20 | 25.26 | 65.33 | 58.67 | 62.00 |
| Average | 57.89 | 66.26 | 29.60 | 67.44 | 60.83 | 64.14 |
| Deepaligned(-GID) | 74.67 | 80.83 | 55.35 | 91.50 | 77.33 | 84.42 |

Table 6: Results on different prompt of DC/GID-DC under IND/OOD=1:1.

(Paraphrase, Verbosity, Simplification) for Chat-GPT (DC/GID-DC) beyond the original prompt and conduct experiments with an IND/OOD ratio of 1:1.[7] The results are shown in Table 6. In summary, different prompts led to slight fluctuations in the experimental outcomes, but the results still support existing conclusions. For example, the results of ChatGPT(DC) are still lower than the baseline Deepaligned.

# 6 Challenge & Future Work

Based on above experiments and analysis, we summarize three challenging scenarios faced by LLMs and provide guidance for the future.

## 6.1 Large scale Clustering

Experiments show that **there are three main reasons why LLMs is limited in performing large-scale clustering tasks:** (1) The maximum length of input tokens limits the number of clusters. (2) When the number of clusters increases, LLMs will have serious recall errors. (3) LLMs have poor robustness to the number of cluster samples.

There have been some work attempts to solve the sequence length constraints of transformer-based models, such as (Bertsch et al., 2023). Another feasible approach is to summarize the samples into topic words before clustering. For recall problem, one post-remediation method is to first screen out the sample index of the recall error after clustering, and then prompt LLMs to complete or delete the sample allocation through multiple rounds of dialogue. For robustness, a possible method is to first estimate the optimal number of cluster, select a small portion of seed samples from the original sample set and cluster them, and then classify the remaining samples into seed clusters.

---

[7]These prompt are also shown in Appendix D.

## 6.2 Semantic understanding of specific domains

In Section 5.2, we find that the main reason for the limited performance of LLMs on GID is the lack of semantic understanding of specific domains. To improve the performance of general LLMs in specific domains, one approach is through fine-tuning LLMs, which often requires high training costs and hardware resources. Another approach is to inject domain knowledge into prompts to enhance LLMs. Section 5.1 and 5.2 show that providing demonstration examples or describing label sets can significantly improve performance, but long prompts will increase the inference cost of each query. **How to efficiently adapt LLMs to specific domains without increasing inference costs** is still an area under exploration.

## 6.3 Cross-domain in-context learning

In some practical scenarios, such as the need to perform a new task or expand business scope, there is often a lack of demonstration examples directly related to the new task. We hope to improve the performance of new tasks by leveraging previous domain demonstrations. However, previous experiments show that cross-domain in-context learning has failed in current LLMs. A meaningful but challenging question is **how in-context learning with IND demonstrations performs well in OOD tasks**? A preliminary idea is to use manual chains of thought to provide inference paths from IND demonstration samples to the labels, thereby producing more fine-grained domain-specific knowledge. These fine-grained intermediate knowledge may help generalize to OOD tasks.

# 7 Conclusion

In this paper, we conduct a comprehensive evaluation of ChatGPT on OOD intent discovery and GID, and summarize the pros and cons of Chat-GPT in these two tasks. Although ChatGPT has made significant improvements in zero or few-shot performance, our experiments show that ChatGPT still lags behind fine-tuned models. In addition, we perform extensive analysis experiments to deeply explore three challenging scenarios faced by LLMs: large-scale clustering, domain-specific understanding, cross-domain in-context learning and provide guidance for future directions.

## Limitations

In this paper, we investigate the advantages, disadvantages and challenges of large language models (LLMs) in open-domain intent discovery and recognition through evaluating ChatGPT on out-of-domain (OOD) intent discovery and generalized intent discovery (GID) tasks. Although we conduct extensive experiments, there are still several directions to be improved: (1) Given the paper's focus on large-scale language models like ChatGPT, it's worth noting that ChatGPT is only accessible for output, which makes it challenging to thoroughly investigate and analyze its internal workings. (2) Although we perform three different data splits for each task, they all come from the same source dataset, which makes their intent granularity consistent. The analysis of different intent granularity is not further explored in this paper. (3) Although we ensure that all experiments on ChatGPT in this paper are based on the same version, further updates of ChatGPT may lead to changes in the results of this paper.

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

## A Datasets

The original dataset Banking contains 77 classes and is class-imbalanced. Its training set has 9003 samples, the validation set has 1000 samples, and the test set has 3080 samples. The average token length of the samples is 11.91 , with the longest being 79. We show 15 IND classes and 15 OOD classes sampled from the 77 class in Table 7.

## B Implementation Details

For all baselines, we use the pre-trained BERT model (bert-base-uncased[8], with 12-layer transformer) as the backbone, and freeze all but the last transformer layer parameters to achieve better performance and speed up the training procedure as suggested in (Zhang et al., 2021). In addition,

---

[8]https://huggingface.com/bert-base-uncased

| IND Class | OOD Class |
|---|---|
| pin_blocked | card_delivery_estimate |
| balance_not_updated_after_bank_transfer | cancel_transfer |
| pending_card_payment | verify_my_identity |
| verify_source_of_funds | cash_withdrawal_charge |
| disposable_card_limits | apple_pay_or_google_pay |
| card_about_to_expire | pending_top_up |
| direct_debit_payment_not_recognised | request_refund |
| top_up_failed | card_linking |
| card_payment_fee_charged | transfer_not_received_by_recipient |
| card_arrival | declined_card_payment |
| card_payment_not_recognised | get_disposable_virtual_card |
| activate_my_card | card_acceptance |
| transfer_timing | get_physical_card |
| getting_spare_card | exchange_rate |
| contactless_not_working | compromised_card |

Table 7: Sampled 15 IND and 15 OOD classes for the experiments. For IND/OOD=3:1 and IND/OOD=3:2, the first 5 and 10 OOD classes are used respectively.

| | OOD Discovery | | | | GID | | |
|---|---|---|---|---|---|---|---|
| Method | ACC | NMI | ARI | Method | IND ACC | OOD ACC | ALL ACC |
| DeepAligned | 94.22 | 95.27 | 90.21 | DeepAligned-GID | 98.67 | 94.22 | 96.44 |
| DKT | 97.78 | 96.97 | 95.16 | E2E | 99.11 | 97.78 | 98.44 |
| ChatGPT(DC) | 80.89 | 84.02 | 62.70 | ChatGPT(GID-DC) | 86.00 | 82.67 | 84.33 |
| ChatGPT(ZSD) | 65.33 | 71.11 | 39.15 | ChatGPT(GID-ZSD) | 88.00 | 72.00 | 80.00 |
| ChatGPT(FSD) | 56.00 | 64.15 | 26.32 | ChatGPT(GID-FSD) | 90.00 | 56.67 | 73.33 |

Table 8: The reults on CLINC under IND/OOD=1:1.

we keep the hyperparameters consistent with those in the official open-source code of all baselines. For ChatGPT-based methods, we perform the experiments by calling OpenAI's official API, and the version of ChatGPT we used is gpt-3.5-turbo-0301. For hyperparameters such as temperature in the API, we keep the default value of OpenAI unchanged.To reduce randomness, we average results over three random runs for all methods of ChatGPT and baselines.

## C Results on the CLINC Dataset

Furthermore, we undertake an exploration of experimental outcomes utilizing an alternative widely employed dataset, CLINC (Larson et al., 2019), in an IND/OOD ratio of 1:1, as presented in Tabel 8. Despite the disparate label granularity and domain between CLINC and Banking Datasets, the experimental results from CLINC align with and bolster the generalizability of our conclusions drawn from the Banking dataset.

## D Details on Prompts

As we perform the GID task on ChatGPT in the manner of a pipeline, we list the complete prompts we used in each stage of the three GID methods in Table 9. In addition, we present the prompt variants of ChatGPT(DC) in Table 10.

## E Cases of Incorrect Recall

We provide cases of incorrect recall in OOD discovery and GID in Figures 11 and 12, respectively.

| Method | Stage | Prompt |
|--------|-------|--------|
| GID-DC | 1 | Next, I will first give you a set of sentences, which will be recorded as Set 1. First, please classify the sentences in Set 1 into 5 (Number of Clusters) categories according to their intentions. You only need to output the category number and the corresponding sentence number in the following format:
Category 1: 1,2,3,4,5 . . . . . .
Then, you need to summarize the intent of each class from Category 1 to Category 5.
Set 1($D_{test}^{OOD}$): 1. can I choose a date for delivery? . . . . . . |
| | 2 | Below is a predefined set of intent categories, recorded as Set 1:1. pin_blocked;. . . . . .
Please classify the following sentence into Set 1 according to its intention, just response the corresponding category number:<test sample> |
| GID-ZSD | 1 | Next, I will first give you a set of intent categories, which will be recorded as Set 1. Then I will give you another set of sentences without intention labels, recorded as Set 2. You first need to learn the knowledge in Set 1, and then use the learned knowledge to classify the sentences in Set 2 into 5 (Number of Clusters) categories according to their intentions. You only need to output the category number and the corresponding sentence number in the following format:
Category 1: 1,2,3,4,5 . . . . . .
It should be noted that the intention in set 1 and the intention in set 2 do not overlap.
Then, you need to summarize the intent of each class from Category 1 to Category 5.
Set 1($Y^{IND}$): pin_blocked; . . . . . .
Set 2($D_{test}^{OOD}$): 1. can I choose a date for delivery? . . . . . . |
| | 2 | Below is a predefined set of intent categories, recorded as Set 1:1. pin_blocked;. . . . . .
Please classify the following sentence into Set 1 according to its intention, just response the corresponding category number:<test sample> |
| GID-FSD | 1 | Next, I will first give you a set of sentences with intention labels, which will be recorded as Set 1.Then I will give you another set of sentences without intention labels, recorded as Set 2. You first need to learn the knowledge in Set 1, and then use the learned knowledge to classify the sentences in Set 2 into 5 (Number of Clusters) categories according to their intentions. You only need to output the category number and the corresponding sentence number in the following format:
Category 1: 1,2,3,4,5 . . . . . .
It should be noted that the intention in set 1 and the intention in set 2 do not overlap.
Then, you need to summarize the intent of each class from Category 1 to Category 5.
Set 1 ($D_{demo}^{IND}$): How do I unblock my card? intention:pin_blocked; . . . . . .
Set 2 ($D_{test}^{OOD}$): 1. can I choose a date for delivery? . . . . . . |
| | 2 | Next, I will first give you a predefined set of intent categories, which will be recorded as Set 1. Then I will give you another set of sentences with intention labels, recorded as Set 2.
Set 1:1. pin_blocked;. . . . . .
Set 2:sentence: How do I unblock my card? intention:pin_blocked;. . . . . .
You first need to learn the knowledge in Set 2, and then use the learned knowledge to classify the following sentence into Set 1 according to its intention , just response the corresponding category : <test sample> |

Table 9: The complete prompts used in the three GID methods.

| Type | Prompt |
|------|--------|
| Original | Next, I will first give you a set of sentences , which will be recorded as Set 1. First,Please classify the sentences in Set 1 into 5 categories according to their intentions. You only need to output the category number and the corresponding sentence number in the following format:
Category 1: 1,2,3,4,5 . . . . . . |
| Paraphrase | I will provide you with a collection of sentences, noted as Set 1. Your task is to categorize the sentences in Set 1 into 5 distinct groups based on their underlying intentions. Your output should include the category number along with the corresponding sentence number, formatted as follows:
Category 1: 1, 2, 3, 4, 5, and so on... |
| Verbosity | Next, I will be presenting you with a compilation of sentences, collectively labeled as "Set 1". Your task is to categorize these sentences into 5 distinct groups according to their underlying intentions. Upon completing the task, your response is anticipated to take the form of a structured enumeration. Your response should consist of the assigned category number along with the respective sentence numbers following this format:
Category 1: 1, 2, 3, 4, 5... |
| Simplification | Next, I'll provide sentences in Set 1. Please categorize them into 5 groups based on intentions. Output the category number and sentence number in this format:
Category 1: 1, 2, 3, 4, 5. . . |

Table 10: The three prompt variants of ChatGPT(DC).

| Input | Response | Missing Recall | Repeated Recall |
|-------|----------|----------------|-----------------|
| Please classify the sentences in Set 2 into 10 categories according to their intentions. | Category 1: 2, 10, 30, 31, 50

Category 2: 4, 25, 27, 34, 44

Category 3: 5, 6, 15, 16, 36

Category 4: 1, 2, 3

Category 5: 7, 17, 27

Category 6: 8, 18, 28, 38

Category 7: 9, 22, 32, 39, 49

Category 8: 11, 12, 21

Category 9: 13, 23, 33, 43

Category 10: 14, 19, 24, 37, 45, 47 | 20,26,29, 35,40,41, 42,46,48 | 2,27 |

Table 11: Cases of incorrect recall in OOD Discovery. The index marked with an underline denotes the repeatedly recalled sample.

| Type | Input | Response | Ground-Truth Intent |
|------|-------|----------|---------------------|
| Missing Recall | How many days until the money will be in my account? | It does not fit in any category in Set 1. | transfer_timing (IND) |
| | I would like to cancel a payment. I purchased something several days ago and i still have not received it. | The sentence does not match any intents in as it is not related to issues with payment or card management. It could potentially fall under a "purchase delivery inquiry" category. | request_refund (OOD) |
| Repeated Recall | I don't know where this transaction came from? | 8. top_up_failed. ( Note: The sentence could also fit under 3. pending_card_payment or 11. card_payment_not_recognised, depending on the context of the conversation.) | card_payment_not_ recognised (IND) |
| | My credit card was declined. | 12. activate_my_card; 24. Issues related to card payme | declined_card_payment (OOD) |

Table 12: Cases of incorrect recall in GID.