# OpenReview forum: "Large Language Models Meet Open-World Intent Discovery and Recognition: An Evaluation of ChatGPT"
_EMNLP/2023/Conference — EMNLP 2023 Main_

### Official Review · Reviewer_x2rM · 2023-08-03

**Soundness:** 3

**Excitement:**

3: Ambivalent: It has merits (e.g., it reports state-of-the-art results, the idea is nice), but there are key weaknesses (e.g., it describes incremental work), and it can significantly benefit from another round of revision. However, I won't object to accepting it if my co-reviewers champion it.

**Paper Topic And Main Contributions:**

This paper investigate the use of ChatGPT on OOD Intent Discovery and General Intent Discovery tasks. OOD Intent Discovery aims to cluster a group of OOD samples into multiple (usually pre-defined number of) categories. Experiments turn out that ChatGPT still underperform fine-tuned methods on both tasks. This paper also list some challenges of ChatGPT on the studied task, and suggest several future directions.

**Questions For The Authors:**

- How are ground-truth labels on OOD samples determined? What criteria is used for this?
- Did you check whether the categorizations by ChatGPT or GPT-4 also made sense and if it was more or less aligned with human preference compared to supervised baselines?

**Reasons To Accept:**

- This paper is the first to study ChatGPT for OOD Intent Discovery and General Intent Discovery task.
- This paper point out several challenges about using ChatGPT for clustering tasks.
- This paper provide comprehensive evaluation results of various setting.

**Reasons To Reject:**

The contributions of this paper are somewhat limited. Firstly, LLMs like ChatGPT are not designed to excel at text clustering tasks. Finding out that ChatGPT does not perform well on clustering is not particularly exciting. Secondly, I remain unconvinced about the usefulness of LLMs for text clustering tasks, because using LLMs on large-scale data clustering is of high computational overhead and low efficiency.  Finally, only very straightforward prompting strategies are investigated in this paper, and only one model of ChatGPT-3.5 is tested, and only one dataset with 5-10 sampled queries are used for evaluation.

**Reproducibility:**

4: Could mostly reproduce the results, but there may be some variation because of sample variance or minor variations in their interpretation of the protocol or method.

**Reviewer Confidence:**

4: Quite sure. I tried to check the important points carefully. It's unlikely, though conceivable, that I missed something that should affect my ratings.

---

> ### Author Rebuttal · Authors · 2023-08-29
>
> We appreciate your thorough review and valuable feedback on our manuscript. We are encouraged by your recognition of the first ChatGPT empirical study on two tasks, the challenges for LLMs in clustering and the comprehensive evaluation in this paper. We sincerely apologize for any unclear presentation and hope this response can resolve your concerns.
>
> Q1: LLMs like ChatGPT aren't designed for text clustering and its poor performance isn't exciting.
>
> A1: We acknowledge that the original design of LLMs such as ChatGPT was not intended for clustering tasks, leading to certain limitations in this regard. However, as general language models, we contend that investigating the capabilities of LLMs in clustering tasks still holds value. Our research has unveiled numerous intriguing phenomena and findings in the realm of clustering, with the subpar performance of ChatGPT in clustering being merely a fractional aspect of the study. Furthermore, in-depth explorations into the reasons behind the underperformance of LLMs is undertaken in Section 5.1 through 5.5, and a comprehensive summary of the challenges faced is provided in Section 6.1, along with guidance for prospective avenues of future research.
>
> Q2: The uselessness of LLMs for text clustering tasks because of high computational overhead and low efficiency.
>
> A2: We understand your concerns regarding the performance and efficiency of LLMs in large-scale data clustering. Indeed, LLMs may incur certain computational overheads when dealing with vast datasets; however, this does not imply that scalability is ineffectual for clustering purposes. Recent studies like [1] and [2] have shown LLMs excel in aiding "user-perspective-based clustering" and "clustering granularity determination." These works also combine LLMs and small models to improve efficiency and performance. Since these are not within the scope of this paper's discussion, it is not elaborated upon herein. The focal point of our research resides in the empirical evaluation of ChatGPT's role in clustering (Section 4.4), a comprehensive analysis of the inherent drawbacks of LLMs in clustering (Section 5.1 to 5.5), and a holistic synthesis of the challenges faced along with proposed solutions (Section 6.1). By addressing these aspects, our endeavor aims to draw attention to overcoming these issues in future research endeavors.
>
> ​	[1] ClusterLLM: Large Language Models as a Guide for Text Clustering
>
> ​	[2] Goal-Driven Explainable Clustering via Language Descriptions
>
> Q3: Only very straightforward prompting, only ChatGPT-3.5 and few samples for testing.
>
> A3: We devise three additional variations (Paraphrase, Verbosity, Simplification) for ChatGPT (DC/GID-DC) beyond the original prompt, and conduct experiments with an IND/OOD ratio of 1:1, as depicted in the results table below:
>
> |                   |       | Discovery |       |         | GID     |         |
> | ----------------- | ----- | --------- | ----- | ------- | ------- | ------- |
> | Prompt/Baseline   | ACC   | NMI       | ARI   | IND ACC | OOD ACC | ALL ACC |
> | Original          | 58.22 | 65.30     | 28.21 | 66.44   | 62.00   | 64.22   |
> | Paraphrase        | 58.67 | 66.80     | 30.10 | 69.33   | 62.00   | 65.67   |
> | verbosity         | 60.89 | 68.74     | 34.85 | 68.67   | 60.67   | 64.67   |
> | Simplification    | 53.78 | 64.20     | 25.26 | 65.33   | 58.67   | 62.00   |
> | Average           | 57.89 | 66.26     | 29.60 | 67.44   | 60.83   | 64.14   |
> | Deepaligned(-GID) | 74.67 | 80.83     | 55.35 | 91.50   | 77.33   | 84.42   |
>
> In summary, different prompts led to slight fluctuations in the experimental outcomes, but the results still support existing conclusions, confirming their robustness.
>
> Regarding the issue pertaining to the number of test models and test samples, we seek to clarify certain misconceptions. The primary subject of investigation in this paper is ChatGPT. While, due to considerations of both space and research focus, we have not presented results for all models in each analytical experiment, we compare and discuss the text-davinci-002, text-davinci-003, and claude models in Section 5.6. As for the query sample quantity, as expounded upon in lines 284-299, owing to the constraints imposed by ChatGPT's input length, we have had to sample 5-10 test samples for each class and a maximum of 30 classes. We deem this to be an acceptable and understandable approach.
>
> Q4: How are ground-truth labels on OOD samples determined? What criteria is used for this?
>
> A4: Following prior work [3] and [4], as well as in accordance with the descriptions in Section 4.1 and Appendix B, we partition a subset of samples from a labeled dataset Banking and utilize them as unlabeled OOD samples by masking their original labels. Consequently, the ground-truth labels for these OOD samples correspond to the original intent labels from the Banking dataset. Furthermore, as indicated in footnote 2, we establish the mapping between the predicted labels and the ground-truth labels using the Hungarian algorithm.
>
> ​	[3] Discovering New Intents with Deep Aligned Clustering
>
> ​	[4] Watch the Neighbors: A Unified K-Nearest Neighbor Contrastive Learning Framework for OOD Intent Discovery
>
> Q5: Did GhatGPT's categories align with human preferences better than supervised baselines?
>
> A5: We appreciate your inspiring insights that comparing LLMs with baselines from a human preference perspective. Based on experimental results, we take into consideration the following two aspects.  In terms of classification accuracy, when utilizing the label distribution of the dataset as a proxy for human preference, ChatGPT performs less effectively than the fine-tuned baselines (Section 4.4).  From the perspective of inductive clustering/categories, supervised baselines can only provide label indexes, whereas LLMs like ChatGPT can generate labels with specific semantics, aligning more closely with human preferences. Additionally, in Section 5.2, we conduct case studies of the quality of labels generated by ChatGPT. Overall, ChatGPT demonstrates remarkable semantic performance while exhibiting a certain level of granularity randomness.  Although the primary focus of this paper is not on human preferences, we acknowledge the significance of human preferences and do not disregard them. We are keen on delving further into this aspect in future endeavors.

---

### Official Review · Reviewer_TNLF · 2023-08-03

**Soundness:** 4

**Excitement:**

4: Strong: This paper deepens the understanding of some phenomenon or lowers the barriers to an existing research direction.

**Paper Topic And Main Contributions:**

The paper mainly focuses on evaluating ChatGPT on out-of-domain (OOD) intent discovery and General Intent Discovery (GID). It proposes a few heuristic prompts using various levels of information from in-domain (IND) data as the context for ChatGPT, and benchmarks these variations against the more standard fine-tuned approaches. The experiments show that ChatGPT significantly outperforms BERT baseline without prior IND knowledge, but is generally weaker than baselines given IND knowledge. The qualitative analyses further dove into potential factors impacting ChatGPT. Lastly, the authors summarize the main challenges with performing clustering tasks with LLMs and give general suggestions to mitigate them.

**Reasons To Accept:**

- The paper claims to be the first to comprehensively evaluate ChatGPT’s performance on OOD intent discovery and GID. Given the popularity of LLMs and the trend in industry to try ChatGPT in any NLP task, such evaluation against strong baselines and calling out the limitations could be very insightful for practitioners.
- The heuristic prompts proposed can serve as reasonable starting points for practitioners who do want to adopt ChatGPT to handle OOD queries.
- The qualitative analyses are informative in showing how sensitive ChatGPT can be w.r.t. prompts, the IND:OOD ratio, and the number of clusters.

**Reasons To Reject:**

- It’d be good to see more analyses and validations on how the prompts from Section 3.2 are engineered. We know that ChatGPT can be very sensitive to prompts. Without seeing sufficient due diligence in prompt engineering, it’s difficult to conclude whether ChatGPT’s relatively poor performance in Section 4.4 is mainly due to suboptimal prompts or inherent limitations of LLMs. Reading Figure 2, it seems that the prompts for ZSD and FSD could potentially mislead ChatGPT into thinking that the test set should have the same set of intents instead of new ones. It could help explain the counter-intuitive result that ChatGPT is doing worse with more IND knowledge.
- Related to the previous point, it’d be good to see more thorough investigations on why ChatGPT is doing worse with more IND knowledge. This topic is briefly covered in Section 5.1, but the analyses are not conclusive to root cause the poor performance, and the statement “current in-context learning of LLMs lacks deep mining and transfer demonstration knowledge capabilities” is too strong and not sufficiently supported (just because the specific transfer learning approach experimented didn’t work it doesn’t mean other approaches won’t).
- The evaluation setting assumes that the OOD test set is composed of only OOD queries with a known number of new intents. It’s a simplified setting compared to real world scenarios where the test set could be a mixture of IND and OOD queries with an unknown number of new intents. Section 5.5 does address the issue of estimating the number of clusters, but the challenges in real world scenarios compared to the experiment setup could be explained earlier.

**Reproducibility:**

4: Could mostly reproduce the results, but there may be some variation because of sample variance or minor variations in their interpretation of the protocol or method.

**Reviewer Confidence:**

4: Quite sure. I tried to check the important points carefully. It's unlikely, though conceivable, that I missed something that should affect my ratings.

**Typos Grammar Style And Presentation Improvements:**

- Line 012 – “extend”
- Table 2, last row – “FSD”
- Line 515 – “an”
- Line 499 – “leading to”
- Line 588 – If “previous experiments” refer to the ones not covered by this paper, you should add some references to support this claim.
- Line 814 – as a nitpick, you can’t completely “avoid randomness” by averaging over a few runs. You can “reduce” or “mitigate the impact of” randomness.

---

> ### Author Rebuttal · Authors · 2023-08-29
>
> We appreciate your thorough review and valuable feedback on our manuscript. We are encouraged by your recognition of the first ChatGPT empirical study on two tasks,  the proposed heuristic prompts and  comprehensive qualitative analyses in this paper. We sincerely apologize for any unclear presentation and hope this response can resolve your concerns.
>
> Q1: More analyses and validations on prompts and the prompts for ZSD and FSD may potentially mislead ChatGPT.
>
> A1: I fully comprehend your concerns. Thorough prompt engineering is crucial to mitigate the variability introduced by different prompts. To address this, we devise three additional variations (Paraphrase, Verbosity, Simplification) for ChatGPT (DC/GID-DC) beyond the original prompt, and conduct experiments with an IND/OOD ratio of 1:1, as depicted in the results table below:
>
> |                   |       | Discovery |       |         | GID     |         |
> | ----------------- | ----- | --------- | ----- | ------- | ------- | ------- |
> | Prompt/Baseline   | ACC   | NMI       | ARI   | IND ACC | OOD ACC | ALL ACC |
> | Original          | 58.22 | 65.30     | 28.21 | 66.44   | 62.00   | 64.22   |
> | Paraphrase        | 58.67 | 66.80     | 30.10 | 69.33   | 62.00   | 65.67   |
> | verbosity         | 60.89 | 68.74     | 34.85 | 68.67   | 60.67   | 64.67   |
> | Simplification    | 53.78 | 64.20     | 25.26 | 65.33   | 58.67   | 62.00   |
> | Average           | 57.89 | 66.26     | 29.60 | 67.44   | 60.83   | 64.14   |
> | Deepaligned(-GID) | 74.67 | 80.83     | 55.35 | 91.50   | 77.33   | 84.42   |
>
> In summary, different prompts led to slight fluctuations in the experimental outcomes, but the results still support existing conclusions, confirming their robustness. Regarding the issue you mentioned about prompts potentially misleading ChatGPT for ZSD and FSD, we extend our sincerest apologies. Throughout the experimental procedure, we explicitly underscored at the terminus of the prompts, "It should be noted that the intention in set 1 and the intention in set 2 do not overlap." Regrettably, this point was inadvertently omitted in Figure 2. We acknowledge this oversight and will correct it in the revised version.
>
> Q2: The analyses are not conclusive to root cause the poor performance and some statement is too strong.
>
> A2: Given the paper's focus on large-scale language models like ChatGPT, it's worth noting that ChatGPT is only accessible for output, which makes it challenging to thoroughly investigate and analyze its internal workings. In Section 4.4(3) and 5.1, we externally analyze the model's output and find that the performance decline is due to "discrepancies in distribution between demonstrations and tasks" and "the treatment of demonstrations as noise". With recent open-sourcing of LLMs like llama2-70B, we're enthusiastic about delving into internal mechanisms for future studies. With regard to the assertion that "current in-context learning of LLMs lacks deep mining and transfer demonstration knowledge capabilities," as you rightly pointed out, different approaches may yield different outcomes. In the revised version, we will employ a more balanced statement, emphasizing potential limitations of the specific transfer learning methods employed in the experiments, while simultaneously acknowledging the potential for success with alternative approaches.
>
> Q3: The challenges in real-world scenarios could be explained earlier.
>
> A3: In order to align with the sequence of previous work[1] [2], we conduct experiments involving a simplified configuration in Section 4.4 and explore more realistic scenarios in Section 5.5. We apologize for any inconvenience caused by the shortcomings in our explanations and will rectify this aspect in the revised version.
>
> [1] Discovering New Intents with Deep Aligned Clustering
>
> [2] Watch the Neighbors: A Unified K-Nearest Neighbor Contrastive Learning Framework for OOD Intent Discovery
>
> Q4: Typos Grammar Style And Presentation Improvements.
>
> A4: Thank you very much for your suggestions. We will enhance our grammar and expression in the revised version based on your feedback.

---

### Official Review · Reviewer_s6Wk · 2023-08-12

**Soundness:** 4

**Excitement:**

3: Ambivalent: It has merits (e.g., it reports state-of-the-art results, the idea is nice), but there are key weaknesses (e.g., it describes incremental work), and it can significantly benefit from another round of revision. However, I won't object to accepting it if my co-reviewers champion it.

**Paper Topic And Main Contributions:**

The paper delves into the evaluation of ChatGPT's capabilities in out-of-domain (OOD) intent discovery and generalized intent discovery (GID). The authors have found that while ChatGPT exhibits decent performance in zero-shot scenarios, it doesn't quite match up to finetuned models. Through a series of analysis experiments, the paper sheds light on the strengths and challenges associated with ChatGPT on these tasks.

**Questions For The Authors:**

A. What's the variance of results across repeated runs in Table 1?

B. In section 4.4 (2) "compare ChatGPT with Finetuned BERT", which rows in Table 1 display these results? There's only one "BERT" row which seems to be under the zero-shot setting.

C. To what extent do your main findings apply to models other than ChatGPT (in Table 5)?

**Reasons To Accept:**

- The paper presents some intriguing, and sometimes counterintuitive, behavioral test observations, e.g. in-domain demonstrations can actually lead to poorer performance than having no demonstrations at all when tested on OOD samples.
- The paper conducts a quite comprehensive qualitative analysis, covering a wide range of aspects, including an ablation study, the effect of LMs,  recall analysis, etc.
- The presentation is clear and relatively easy to follow.

**Reasons To Reject:**

- While the paper presents some interesting behavioral test observations, I would love to see a deeper exploration into the underlying reasons for them. For instance, why do in-domain demonstrations result in subpar performance compared to no demonstrations? The authors provide conjectures for some of these observations, but didn't verify them. Therefore, the current takeaways of the paper look more on the shallower side and don't really look particularly insightful.
- There is no variation of prompt phrasing and exemplar choice. It will be interesting to see how generalizable the current findings are to such variations.
- Only a single dataset is used in evaluation, which may raise questions about the generalizability of the findings.
- No code or data is provided, which can make it hard to reproduce the results.

**Reproducibility:**

2: Would be hard pressed to reproduce the results. The contribution depends on data that are simply not available outside the author's institution or consortium; not enough details are provided.

**Reviewer Confidence:**

4: Quite sure. I tried to check the important points carefully. It's unlikely, though conceivable, that I missed something that should affect my ratings.

**Typos Grammar Style And Presentation Improvements:**

- Some parts of the paper reads a bit too dry, like a technical report, especially when the use of acronyms is too dense (like in Section 4.4 (3)). You could just call them "zero-shot" and "few-shot" instead of using the acronyms.

- It would be good if Figures 4 and 5 can be made colorblind-friendly.

- Line 8: "some studies has been ..."

- Line 496: "ChatGPT is difficult to discover ..." -> "It's difficult for ChatGPT to discover ..."

---

> ### Author Rebuttal · Authors · 2023-08-29
>
> We appreciate your thorough review and valuable feedback on our manuscript. We are encouraged by your recognition, especially since you believe that this paper presents some intriguing experimental observations, comprehensive qualitative analysis, and clear writing. We sincerely apologize for any unclear presentation and hope this response can resolve your concerns.
>
> Q1: The authors provide conjectures for some of these observations, but didn't verify them.
>
> A1: Given the paper's focus on large-scale language models like ChatGPT, it should be noted that ChatGPT is only available for output, which makes it challenging to thoroughly investigate and analyze its internal workings. Nevertheless, we are committed to conducting experiments and  analysis from an external perspective. We have put forth our best efforts to design experiments rigorously and to seek possible explanations by comparing outputs in different scenarios, aiming to understand the reasons behind the model's behavior. For instance, regarding the issue of in-domain demonstrations leading to lower performance, we provide an initial explanation in section 4.4(3) and  further summarize underlying reasons through comparative experiments and connections to prior research [1] in section 5.1. We fully comprehend your expectations for more in-depth explanations. With recent open-sourcing of LLMs like llama2-70B, we're enthusiastic about delving into internal mechanisms for future studies.
>
> [1] Rethinking the role of demonstrations: What makes in-context learning work?
>
> Q2: There is no variation of prompt phrasing and exemplar choice.
>
> A2: We value your feedback and recognize that introducing these variations can further enhance the generalizability of the research findings. We devise three additional variations (Paraphrase, Verbosity, Simplification) for ChatGPT (DC/GID-DC) beyond the original prompt, and conduct experiments with an IND/OOD ratio of 1:1, as depicted in the results table below:
>
> |                   |       | Discovery |       |         | GID     |         |
> | ----------------- | ----- | --------- | ----- | ------- | ------- | ------- |
> | Prompt/Baseline   | ACC   | NMI       | ARI   | IND ACC | OOD ACC | ALL ACC |
> | Original          | 58.22 | 65.30     | 28.21 | 66.44   | 62.00   | 64.22   |
> | Paraphrase        | 58.67 | 66.80     | 30.10 | 69.33   | 62.00   | 65.67   |
> | verbosity         | 60.89 | 68.74     | 34.85 | 68.67   | 60.67   | 64.67   |
> | Simplification    | 53.78 | 64.20     | 25.26 | 65.33   | 58.67   | 62.00   |
> | Average           | 57.89 | 66.26     | 29.60 | 67.44   | 60.83   | 64.14   |
> | Deepaligned(-GID) | 74.67 | 80.83     | 55.35 | 91.50   | 77.33   | 84.42   |
>
> On the whole, although the introduction of variations in prompt phrasing induces slight fluctuations in ChatGPT's performance, our conclusions remain robust, including observations such as "ChatGPT underperforms compared to the baseline." Regarding exemplar choice, we examine the impact of different demonstration strategies in Section 5.1 and Figure 4. We will present the design of these prompt variations in the revised version.
>
> Q3: Only a single dataset is used in evaluation.
>
> A3: We recognize concerns about the universality of research outcomes from a single dataset. To address this, we undertake an exploration of experimental outcomes utilizing an alternative widely employed dataset, CLINC[2], in an IND/OOD ratio of 1:1, as presented below:
>
> |              |       | Discovery |       |                  |         | GID     |         |
> | ------------ | ----- | --------- | ----- | ---------------- | ------- | ------- | ------- |
> | Method       | ACC   | NMI       | ARI   | Method           | IND ACC | OOD ACC | ALL ACC |
> | DeepAligned  | 94.22 | 95.27     | 90.21 | DeepAligned-GID  | 98.67   | 94.22   | 96.44   |
> | DKT          | 97.78 | 96.97     | 95.16 | E2E              | 99.11   | 97.78   | 98.44   |
> | ChatGPT(DC)  | 80.89 | 84.02     | 62.70 | ChatGPT(GID-DC)  | 86.00   | 82.67   | 84.33   |
> | ChatGPT(ZSD) | 65.33 | 71.11     | 39.15 | ChatGPT(GID-ZSD) | 88.00   | 72.00   | 80.00   |
> | ChatGPT(FSD) | 56.00 | 64.15     | 26.32 | ChatGPT(GID-FSD) | 90.00   | 56.67   | 73.33   |
>
> Despite the disparate label granularity and domain between CLINC and Banking datasets, the experimental results from CLINC align with and bolster the generalizability of our conclusions drawn from the Banking dataset.
>
> [2] An Evaluation Dataset for Intent Classification and Out-of-Scope Prediction
>
> Q4: No code or data is provided.
>
> A4: Although key information regarding the reproducibility of our results is presented in Figure 2  and Appendices A, B, and D of the paper, we sincerely apologize for not furnishing a more extensive compilation of data and code support. We hereby commit to supplying a comprehensive set of data and code subsequent to the conclusion of the blind review process, thereby ensuring facile reproducibility of our experimental outcomes by fellow researchers.
>
> Q5: The variance of results in Table 1.
>
> A5: In Table 1, we conduct three repeated experiments to mitigate the errors induced by randomness. The table below presents Table 1 with variance values:
>
> |              |              | IND:OOD=3:1  |              |              | IND:OOD=3:2 |             |             | IND:OOD=1:1 |             |
> | ------------ | ------------ | ------------ | ------------ | ------------ | ----------- | ----------- | ----------- | ----------- | ----------- |
> | Method       | ACC          | NMI          | ARI          | ACC          | NMI         | ARI         | ACC         | NMI         | ARI         |
> | BERT         | 52(0)        | 41.58(0)     | 15.36(0)     | 36(0.89)     | 42.33(0.99) | 5.616(0.95) | 29.33(0)    | 45.26(0.31) | 2.499(0.19) |
> | ChatGPT(DC)  | 88(3.27)     | 84.62(5.04)  | 73.36(7.33)  | 78(3.27)     | 78.2(2.54)  | 55.32(5.58) | 58.22(2.26) | 65.3(0.71)  | 28.21(1.83) |
> | Deepaligned  | 100(0)       | 100(0)       | 100(0)       | 78.67(0.94)  | 82.18(0.93) | 61.01(2.08) | 74.67(0)    | 80.83(0.14) | 55.35(0.64) |
> | DKT          | 93.33(2.49)  | 91.71(5.52)  | 84.82(2.44)  | 80.67(0.94)  | 82.59(1.93) | 64.53(3.79) | 76.45(1.66) | 83.23(1.49) | 60.92(2.25) |
> | ChatGPT(ZSD) | 92(5.66)     | 87.25(9.02)  | 80.16(12.21) | 67.33( 3.40) | 68.72(2.81) | 39.32(8.00) | 50.67(6.06) | 60.73(5.19) | 21.25(8.17) |
> | ChatGPT(FSD) | 74.67(10.50) | 64.77(11.62) | 45.92(13.73) | 56.67(3.40)  | 63.56(2.30) | 31.47(3.98) | 49.78(1.26) | 60.98(0.91) | 20.72(1.12) |
>
> We observe that the ChatGPT methods exhibit higher variance compared to the baselines. This phenomenon could be attributed to the inherent characteristics of language generation in LLMs.
>
> Q6: Which rows in Table 1 display the “Finetuned BERT” results?
>
> A6: The term 'fine-tuning BERT' refers to two baselines with IND prior, namely DKT and DeepAligned.
>
> Q7: The extensibility of findings in other models.
>
> A7: We appreciate and value your questions regarding the extensibility of our findings to other models. Given our paper's primary focus on ChatGPT, we omitted some model results in certain experiments due to space and research priorities. However, as one of the forefront advanced large-scale models currently available, ChatGPT to a significant extent can serve as a representative of contemporaneous LLMs. Additionally, we present and analyze the performance of text-davinci-002, text-davinci-003, and claude models in Section 5.6 and Table 5. In forthcoming revisions, we intend to supplement the experimental outcomes of additional models in the appendix, thereby ensuring the comprehensiveness and applicability of the study.
>
> Q8: Typos Grammar Style And Presentation Improvements.
>
> A8: We will enhance our grammar and expression in the revised version based on your feedback.

---

### Meta-Review · Area_Chair_6QAM · 2023-09-19

**Recommendation:** 4

**Metareview:**

The paper evaluates ChatGPT for the tasks of out-of-domain intent discovery and generalised intent discovery. They show that while ChatGPT performs well in zero-shot settings. However, it still does not outperform finetuned models.

The reviewers were positive about this work and appreciated that this is the first work to evaluate ChatGPT's capabilities in out-of-domain (OOD) intent discovery and generalized intent discovery (GID). The following concerns were raised by the reviewers:

1. Lack of variations in prompts which may affect the performance of ChatGPT - the authors address this concern partially in the rebuttal and try experiment with more variations in the prompts.
2. Only one dataset was used - the authors have addressed this by providing additional results on other datasets
3. No thorough analysis of the results and no investigation on why ChatGPTs' performance is poor - the authors mention this is beyond the scope of the paper but I do not fully agree.
4. Models other than ChatGPT have not been tried - Given that CHatGPT was state of the art at the time this work was undertaken and considering the costs of experiments with multiple LLMs I think it okay to keep the scope limited to GPT.

Reviewer x2rM had raised some other concerns but those have been clarified.

Overall, I find this paper to be interesting and worth publishing. I would request the authors to address the following in the final version:

1. Add all additional results shared during the rebuttal phase to the main body or appendix of the final paper, as appropriate.
2. Honor their commitment made during the response period of making the code, data and evaluation scripts publicly available.
3. Address some of the concerns related to investigating the reason for poor performance.

---

### Decision · Program_Chairs · 2023-10-07

**Decision:**

Accept-Main

**Comment:**

The paper evaluates ChatGPT for the tasks of out-of-domain intent discovery and generalised intent discovery. They show that while ChatGPT performs well in zero-shot settings. However, it still does not outperform finetuned models.

The reviewers were positive about this work and appreciated that this is the first work to evaluate ChatGPT's capabilities in out-of-domain (OOD) intent discovery and generalized intent discovery (GID). The following concerns were raised by the reviewers:

1. Lack of variations in prompts which may affect the performance of ChatGPT - the authors address this concern partially in the rebuttal and try experiment with more variations in the prompts.
2. Only one dataset was used - the authors have addressed this by providing additional results on other datasets
3. No thorough analysis of the results and no investigation on why ChatGPTs' performance is poor - the authors mention this is beyond the scope of the paper but I do not fully agree.
4. Models other than ChatGPT have not been tried - Given that CHatGPT was state of the art at the time this work was undertaken and considering the costs of experiments with multiple LLMs I think it okay to keep the scope limited to GPT.

Reviewer x2rM had raised some other concerns but those have been clarified.

Overall, I find this paper to be interesting and worth publishing. I would request the authors to address the following in the final version:

1. Add all additional results shared during the rebuttal phase to the main body or appendix of the final paper, as appropriate.
2. Honor their commitment made during the response period of making the code, data and evaluation scripts publicly available.
3. Address some of the concerns related to investigating the reason for poor performance.